# Drought characteristics and their impact on vegetation net primary productivity in the climate-sensitive transition zones of North China

Xueying Zhang, Bo Zhang◉*, Xiao Hou, Di An, Zhexi Wu

College of Geography and Environmental Science, Northwest Normal University, Lanzhou, China

* zhangbo@nwnu.edu.cn

## Abstract

Net Primary Productivity (NPP) serves as a key measure of ecosystem production capacity and carbon absorption ability. Evaluating the arid characteristics and their NPP impact in the climate-sensitive transition zones of North China (North China's humid and semi-humid warm temperate regions) is vital for studying regional climate change, improving ecosystem adaptation, and promoting sustainable development. We analyze drought characteristics at multiple temporal scales (2001−2020) using the Standardized Potential Evapotranspiration Index (SPEI) and Standardized Soil Moisture Index (SSMI). The Carnegie-Ames-Stanford approach (CASA) model was employed to simulate and estimate NPP values, quantify NPP dynamics, and assess vegetation productivity responses to drought stress. We found that droughts showed a slight easing trend from 2001 to 2020. Over the long term, drought intensity and extent were relatively minor. Over the short term, there were frequent occurrences of short-duration, high-intensity droughts. Overall, NPP showed an upward trend, with a decreasing spatial distribution from the central and north-central regions toward the east and west. The fastest average annual growth rate was observed in grassland NPP (8.02g C·m$^{-2}$a$^{-1}$), followed by woodland (4.03g C·m$^{-2}$a$^{-1}$). Precipitation is an important factor that affects vegetation NPP. Areas with a high correlation between NPP and SPEI-12 are primarily grasslands. Significant relationships exist between NPP and seasonal SPEI-3: positive in summer and negative in winter. Grasslands are most sensitive to drought; forests, on the other hand, exhibit drought resistance and a delayed response to such conditions.

## 1. Introduction

Vegetation plays a critical role in terrestrial ecosystems and the carbon cycle, serving as a responsive gauge of global environmental change [1]. Net primary productivity (NPP), quantified as photosynthetic production minus plant respiration per unit time and space [2], is fundamental to gauge ecosystem stability, service provision, and

**Data availability statement:** The relevant data on which the research results described in this manuscript are based, as well as the minimal data set required to reproduce our research results, are included in our manuscript (the minimal data set required to reproduce our results mainly includes meteorological, MODIS, and SPEI data). In addition, meteorological, MODIS, and SPEI data are all third-party data, to which the authors do not have special access; other researchers can access these data in the same way as the authors. These data can be obtained as follows: 1. Meteorological data can be obtained from https://data.tpdc.ac.cn/ and https://www.ecmwf.int/en/era5-land. 2. MODIS NPP and land cover products can be obtained from https://earthengine.google.com. 3. SPEI products can be obtained from https://www.scidb.cn.

**Funding:** This research was funded by the National Natural Science Foundation of China (41561024).

**Competing interests:** The authors have declared that no competing interests exist.

recovery capabilities. Remote sensing enables large-scale NPP monitoring, making it the mainstream estimation method. Current models include climatic productivity, ecological process, and light energy utilization types [3]. Remote sensing-based light energy utilization models (e.g., CASA, GLO-PEM) excel in regional NPP monitoring through parameters such as vegetation-absorbed photosynthetically active radiation (FPAR) and light use efficiency [4]. Specifically, the CASA model demonstrates wide applicability for NPP estimation [5]. Zhu et al. reduced arid zone inversion errors by 18% through increasing cold-arid grassland $\varepsilon_{max}$ from 0.16 to 0.23 [6]. DeFries et al. applied CASA to quantify anthropogenic land cover impacts on global carbon emissions and NPP [7]. Sun et al. compared three models—the climate-based model, synthetic model, and CASA model—and concluded that the CASA model is more suitable for simulating NPP across China [8].

Drought, a long-cycle, large-scale meteorological disaster, threatens ecological and socioeconomic stability through chain reactions like water scarcity and crop loss. In China (1960–2014), drought severity intensified significantly in northern, southwestern, and southern regions, with northern China experiencing 45 years of severe drought, including frequent extremes [9]. Drought affects vegetation bidirectionally: short-term photosynthetic suppression via stomatal closure and long-term community restructuring [10](e.g., increased short-lived plants in Central Asian grasslands). Impacts vary spatiotemporally: global extreme droughts (1/100-year) cause ≤35% NPP losses, especially in semi-arid zones [11]. The intensification or alleviation of aridity in the Loess Plateau region contributes differently to vegetation NPP [12]. In northeast China's forests, severe droughts reduced annual NPP by 11.09 Tg C, with Daxing'anling suffering more than Changbai Mountain [13]. In 64.72% of North China's arid zones, GPP increased significantly under water-limited conditions [14]. Common drought indices (e.g., Standard Precipitation Index (SPI), Palmer Drought Index (PDSI), SPEI) were applied effectively, as demonstrated by Čadro et al. using SPEI to quantify drought severity, amplitude, and duration in Bosnia and Herzegovina [15]. SPEI effectively monitors drought globally, including in Turkey's Euphrates Basin [16], in a global context, China as a whole [17], Northeast China [18], Northwest China [19], and the North China Plain [20]. It outperforms other indices in climate change response sensitivity, enhancing drought intensity/frequency assessment. The Standardized Soil Moisture Index (SSMI) is an agricultural drought monitoring indicator constructed based on historical soil moisture time series. This method not only features a relatively straightforward calculation process but also accounts for the distributional characteristics of soil moisture data. Research on its applicability in regional agricultural drought monitoring provides a scientific basis for establishing operational drought monitoring systems and assessing their impacts [36]. SPEI can provide early warnings of climate risks associated with water deficits, while SSMI can verify whether drought has already caused substantive impacts on surface ecosystems and agricultural production. The two indices form a highly complementary relationship in terms of physical mechanisms, spatiotemporal attributes, and monitoring targets. The combined use of SPEI and SSMI overcomes the limitations of individual indices, providing a more systematic, multidimensional, and reliable scientific

framework for drought early warning, precise assessment, process tracking, and impact analysis. Therefore, this study employs the SPEI index to analyze drought characteristics from 2001 to 2020, assess their impact on NPP, and utilize the SSMI to validate the substantive effects of drought on NPP in terrestrial ecosystems.

This study focuses on North China's humid and semi-humid warm temperate regions—a climatically sensitive transition between humid and arid climates with complex subsurface and diverse vegetation [21]. This region falls within the warm temperate climate zone, characterized by warm temperate deciduous broadleaf forests as its typical zonal vegetation. It occupies a classic transitional belt shifting from humid to arid and semi-arid conditions. The delineation of this area primarily follows the natural geographical zoning scheme for China proposed by Zhao [22]. Yang et al. found that between 1951 and 1999, the fluctuation range of China's wet-dry climate boundary in the North China region reached 40–400 kilometers, significantly higher than in other areas, indicating this region's sensitivity to environmental changes [23]. From 1961 to 2014, the semi-arid zone in the North China Plain expanded continuously, with pronounced aridification trends particularly evident in Tianjin and eastern Hebei. Conversely, parts of Henan and Shandong exhibited slight humidification due to changes in evapotranspiration [24]. These complex shifts in wet-dry patterns are characteristic features of transitional zones. Therefore, this study designates the humid and semi-humid warm temperate zone of North China as the research area for the " climate-sensitive transition zones of North China." Selecting this region as representative of the climate-sensitive transition zones of North China aims to investigate the impact of arid stress on ecosystem productivity using a typical unit featuring a complete thermohygric gradient and clear ecological boundaries. This zoning scheme and conceptual definition facilitate understanding the essence of the research question within the macro-ecological geographical pattern. Meanwhile, considering the degree gradient, temporal length, and variability of drought's influence on vegetation NPP, the interaction between the two calls for further investigation. We analyze multi-scale drought patterns (1/3/12-month SPEI) coupled with CASA-modeled NPP dynamics (2001–2020) to quantify drought impacts on vegetation NPP across spatiotemporal gradients. Results provide quantitative support for regional ecological resilience assessment, drought adaptation, and dual-carbon targets.

## 2. Materials and methods

### 2.1. Study area

North China's humid and semi-humid warm temperate regions are bounded by the ≥ 10°C cumulative temperature 3200°C isotherm (range 3,200–4,500°C, frost-free period 150–200 days) [22]. Its geographical scope extends south to the Qinling Mountains and Huai River, west to the Wushaoling-Qilian Mountains, and covers Beijing, Tianjin, the entire province of Shandong, most of Hebei, Shanxi, Shaanxi, and Henan, as well as parts of Gansu, Ningxia, Anhui, and Jiangsu (Fig 1a). The climate is distinguished by annual precipitation of 400–800 mm and concentrated summer showers, and the geography

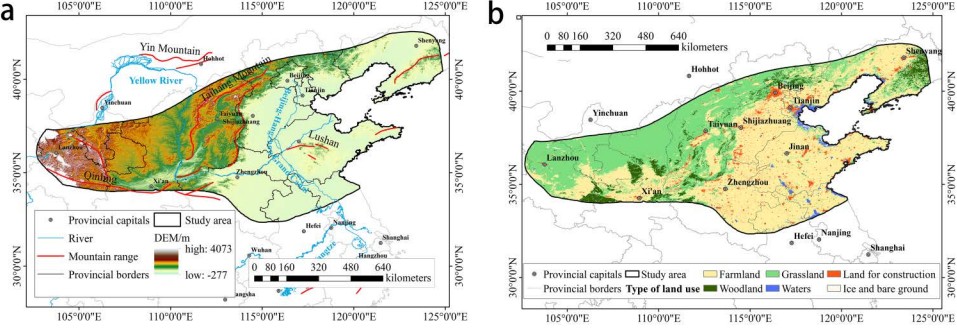

**Fig 1. a) Location and elevation of the study area; b) Land cover map of the study area.** (map lines delineate study areas and do not necessarily depict accepted national boundaries).

differs between the mountainous northwest (grassland or woodland dominated) and the lowlands in the southeast (primary agricultural production areas).

## 2.2. Data sources

**2.2.1. Meteorological data.** This study utilized meteorological data (average temperature [25] and monthly precipitation [26] for China, 1901–2023), and an aridity index (AI) dataset (1 km resolution, China, 1901–2024) from Peng [27]. All data were sourced from the Qinghai-Tibet Plateau Scientific Data Center (https://data.tpdc.ac.cn/). Using ArcMap 10.8, monthly precipitation, temperature, and AI data for the study area (2001–2020) were extracted and cropped using a mask.

Total solar radiation data were sourced from the European Center for Medium-Range Weather Forecasts (ECMWF) ERA5-Land dataset [28] (https://www.ecmwf.int/en/era5-land), originally at about 9 km spatial resolution. Spatial interpolation was then applied to achieve a 1 km resolution.

**2.2.2. Soil moisture data.** This study acquired the global daily surface soil moisture dataset (1 km spatial resolution, 2000–2020) from the Qinghai-Tibetan Plateau Scientific Data Center (https://data.tpdc.ac.cn/) [29]. Using a mask in Arc Map 10.8, the data were extracted and cropped, focusing on surface soil moisture within the study area for the period 2001–2020.

**2.2.3. Land use data.** Utilizing the MODIS MCD12Q1 dataset (500 m resolution) [30] accessed through Google Earth Engine (GEE) (https://earthengine.google.com), we acquired 2001–2020 land use data for the target region. This imagery was downloaded, reprocessed to 1 km resolution, and reclassified into six land cover types following the IGBP scheme: farmland, Woodland, grassland, water, land for construction, and ice and bare ground. (Fig 1b).

**2.2.4. SPEI.** The SPEI data originated from a 1 km resolution dataset focused on drought monitoring and impact assessment across mainland China (2001−2020), accessed via the Science Data Bank (https://www.scidb.cn) [31]. SPEI-1 is derived from monthly water balance data that is non-cumulative, fitted to a probability distribution, and standardized. It represents the monthly water balance deficit or surplus. This index measures very short-term, near-real-time meteorological drought conditions. It is primarily controlled by monthly precipitation and temperature (which influence evapotranspiration). SPEI-3 is derived from a three-month sliding cumulative monthly water balance, followed by probability distribution fitting and normalization. It represents the cumulative water balance over the past three months (including the current month). It is a seasonal-scale indicator reflecting seasonal moisture trends. SPEI-12 is derived by sliding and accumulating monthly water balances over 12 months, followed by probability distribution fitting and normalization. It represents the cumulative water balance deficit over the past 12 months (including the current month). As an annual-scale indicator, it reflects long-term hydrometeorological trends. Table 1 presents the SPEI-based classification scheme for drought severity levels.

**2.2.5 Remote sensing data.** The MOD17A3HGFv061 product served as the primary data source for net primary productivity (NPP) estimates at 1-km spatial resolution, accessed via the Google Earth Engine (GEE) platform (https://earthengine.google.com) [32]. Complementary Normalized Difference Vegetation Index (NDVI) observations were derived

**Table 1. Grade standard of drought.**

| SPEI Range | Drought Level |
|---|---|
| −0.5 < SPEI | No drought |
| −1 < SPEI ≤ −0.5 | Mild drought |
| −1.5 < SPEI ≤ −1 | Moderate drought |
| −2 < SPEI ≤ −1.5 | Severe drought |
| SPEI ≤ −2 | Extreme drought |

 

from the MOD13A1v061 dataset, initially at 500-m resolution on the identical GEE infrastructure. To ensure dimensional consistency with NPP data, these NDVI layers underwent resampling processing to achieve a unified 1-km grid scale.

### 2.3. Data processing methods

**2.3.1. CASA model simulation of NPP.** As a model simulating photosynthetic light use efficiency, the CASA framework was employed in this study to derive both annual and monthly vegetation NPP values across the North China's humid and semi-humid warm temperate regions. The model's foundational algorithms are:

$$NPP(x, t) = APAR(x, t) \times \epsilon(x, t) \tag{1}$$

$$APAR(x, t) = 0.5SOL(x, t) \times FPAR(x, t) \tag{2}$$

Where $APAR(x, t)$ denotes the photosynthetically active radiation absorbed (MJ·m$^{-2}$) by vegetation at location $x$ during temporal interval $t$; the prevailing light use efficiency $\varepsilon(x, t)$ is derived from temperature/moisture stress coefficients combined with maximum light use efficiency [33]; the coefficient 0.5 quantifies the ratio of photosynthetically active radiation relative to total solar radiation; $SOL(x, t)$ corresponds to total solar radiation; and $FPAR(x, t)$ quantifies the fractional absorption of photosynthetically active radiation by vegetation, with its magnitude derived from NDVI datasets. $FPAR(x, t)$ shows how much of the light that plants can use is absorbed by vegetation, and we can estimate FPAR using data from the normalized vegetation index.

To compute the monthly NPP, this study employed Zhu's modified CASA model [6], which was run in ENVI software, and input data such as monthly rainfall, average monthly temperature, monthly sunshine, NDVI, and land use type from North China's humid and semi-humid warm temperate regions.

**2.3.2. Dry-wet changes and drought identification.** The Aridity Index (AI) is a core regional aridity assessment metric [34], especially applicable to North China with its climate-driven aridity fluctuations. AI captures precipitation-evaporation dynamics and delineates aridity gradients and environmental sensitivity. We therefore employ AI to analyze aridity changes in North China's humid and semi-humid warm temperate regions. AI classification: Humid (AI < 1), Semi-humid (1 ≤ AI < 1.5), Semi-arid (1.5 ≤ AI < 4), Arid (AI ≥ 4).

The drought frequency index (DFI) measures the ratio of drought months to total months during the study period. Higher DFI values indicate greater drought frequency [35]. When SPEI ≤ −1, it is considered that a "drought" has occurred. The formula for calculating the frequency of droughts is given in Equation (3):

$$DFI = \left(\frac{m}{M}\right) \times 100\% \tag{3}$$

Where DFI represents the frequency of droughts; m denotes the number of months during which droughts occur; M represents the total number of months within this time scale.

The SSMI is a standardized measure of soil moisture [36]. The formula for constructing the SSMI is shown in Equation (3):

$$SSMI = \frac{SM - \overline{SM}}{\sigma} \tag{4}$$

Where $SM$ refers to instantaneous soil moisture for a given temporal scale; $\overline{SM}$ represents the corresponding multi-year mean soil moisture, and $\sigma$ quantifies its multi-year standard deviation at that identical scale. The drought class was divided according to the SSMI index, as shown in Table 2.

**Table 2. SSMI grade standard of drought.**

| SSMI Range | Drought Level |
|---|---|
| 0 < SSMI | No drought |
| −1 < SSMI ≤ 0 | Mild drought |
| −1.5 < SSMI ≤ −1 | Moderate drought |
| −2 < SSMI ≤ −1.5 | Severe drought |
| SSMI ≤ −2 | Extreme drought |

**2.3.3. Statistical analysis.** We applied Theil-Sen median trend analysis with Mann-Kendall (M-K) testing to assess temperature, precipitation, SPEI, SSMI, and NPP temporal trends. This robust approach minimizes outlier impacts [37–38], with the distribution-free M-K test accommodating non-normal data [39].

Statistical linkages between vegetation NPP, drought severity (SPEI), and key climatic drivers—including air temperature, precipitation, and solar radiation—were assessed via Pearson correlation analysis to quantify their ecophysiological associations [40]. Partial correlation analysis measured NPP-climate relationships with t-tests verifying significance [41].

## 3. Results and analysis

### 3.1. Dry and wet changes and drought identification

Fig 2a shows that the mean value of temperature in the study area during 2001–2020 was 10.74°C, with an increasing trend (0.02°C/a), reaching a peak of 11.26°C in 2019 and a low value of 10.03°C in 2012. The mean value of precipitation for the same period was 580.52 mm, increasing at a rate of 2.65 mm/a, with a maximum of 717.38 mm in 2003 and a minimum of 481.98 mm in 2002. AI analysis shows decreased humid and arid areas but increased semi-humid and semi-arid zones in the study region.

By comparing selected historical drought events with SPEI values from the 2001–2021 *China Meteorological Disaster Yearbook*, we calculated the Probability of Detection (POD = 0.6957), False Alarm Ratio (FAR = 0.1579), and Critical Success Index (CSI = 0.7273). This confirms that SPEI effectively identifies historical drought events in the humid and semi-humid temperate regions of North China, demonstrating its practicality for drought monitoring. The correlation between SPEI and SSMI is extremely high (the positive correlation reaches 91.21%, and in 54.61% of the regions, this correlation is statistically significant). SPEI-12 indicates inter-annual drought, while SSMI characterizes inter-annual soil

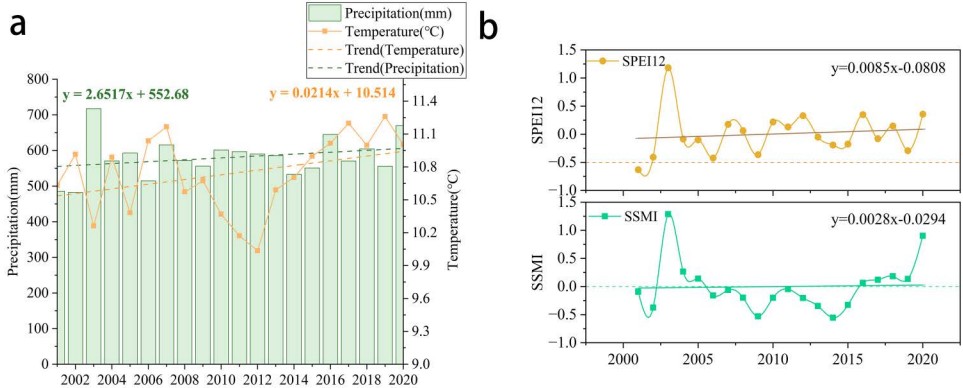

**Fig 2. a) Interannual dynamics of precipitation and mean temperature (2001-2020); b) Temporal evolution of drought indices (SPEI-12 vs. SSMI) (2001-2020).**

moisture, and the trajectories of the two changes are nearly identical (Fig 2b).The study found that SPEI-12 increased by 0.0085/a annually, peaking at 1.18 in 2003 (minimum (−0.63) in 2001), indicating that the region was dominated by no drought and mild drought, with slight drought mitigation, whereas the SSMI changed by 0.0028/a annually, peaking at 1.29 in 2003 (minimum −0.55 in 2014), indicating mild drought status.

SPEI-1 showed mild drought affected 17.04% of the area, moderate drought affected 11.41%, and extreme drought (0.63%) was concentrated in the 2010–2014 winter/spring (Fig 3a), peaking at 22.36% in December 2010. Conversely, SPEI-12 was dominated by light drought (16.96%), with extreme drought at 0.004% (Fig 3b). SPEI-3 revealed multi-season drought extremes during 2001−2015. Trend analysis indicated the SPEI-1 drought area increased (0.015/a) while the SPEI-12 decreased (−1.29/a), reflecting frequent short-term acute droughts. Seasonally, drought extent rose in winter (0.42/a) but declined in spring (−0.3/a), summer (−0.95/a), and autumn (−0.49/a).

Monthly (Fig 4a) shows 92.27% of the area with 30–35% drought frequency, peaking over 38% in southern Hebei and northern Henan; yearly (Fig 4b) reveals high-frequency areas (38% and 40%) in western and northeastern parts; season-ally (Fig 4c) displays spring droughts peaking at 48.13%, while summer and winter exhibit opposite east-west patterns. This spatial pattern reflects monsoon and vegetation differences: summer brings eastern coasts higher precipitation and farmland coverage, reducing drought, contrasting with western areas intensified by Taihang Mountains' moisture blocking [42]; winter decreases eastern precipitation and vegetation cover, elevating drought frequency.

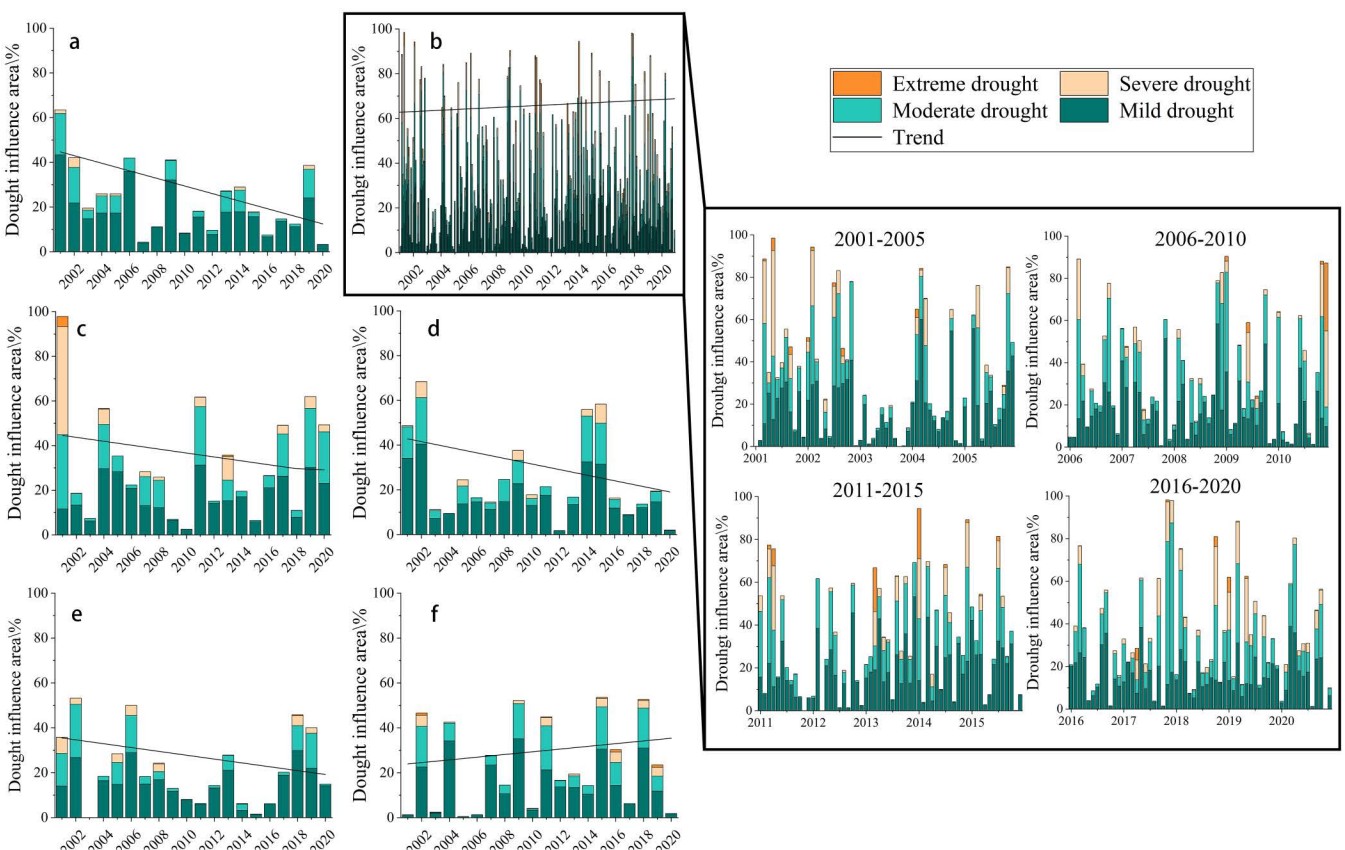

**Fig 3. Area of influence of drought severity by time scale from 2001-2020: a) year; b) month; c) spring; d) summer; e) fall; f) winter.**

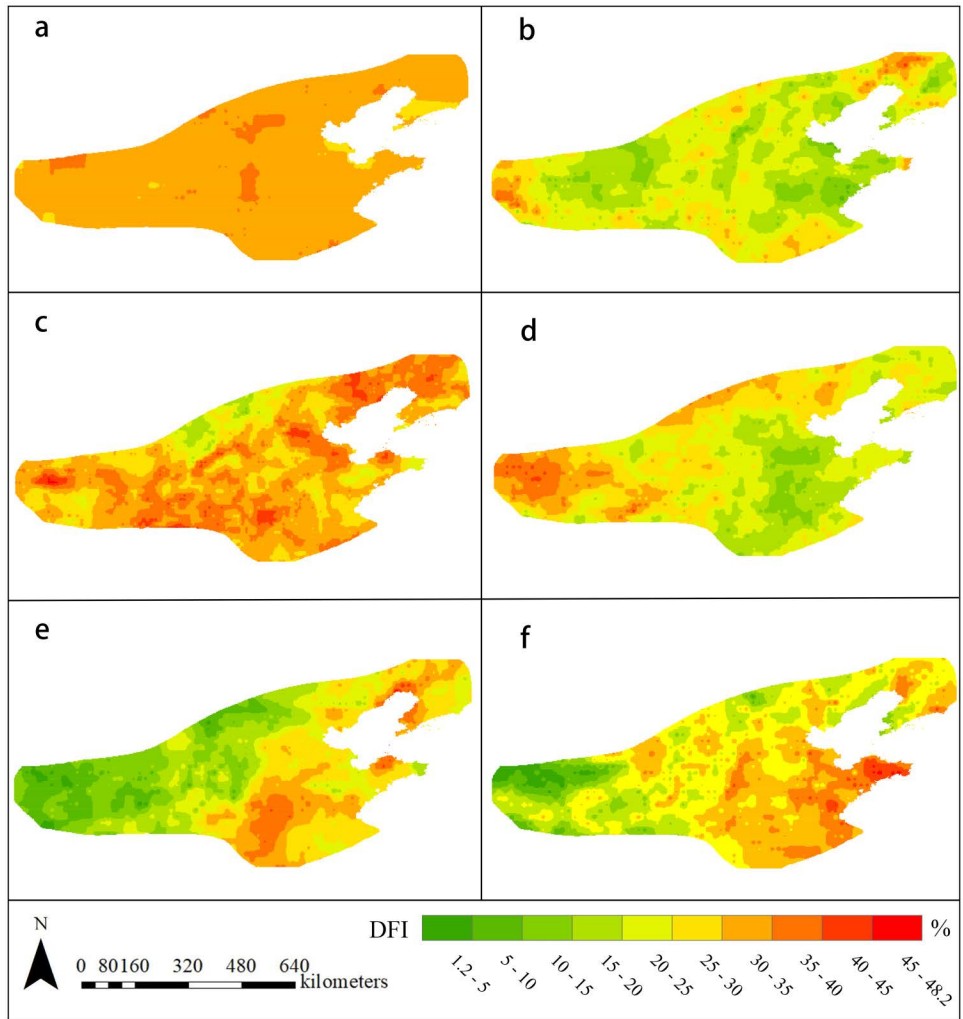

**Fig 4. Drought Frequency Index (DFI), 2001−2020 a) SPEI-1; b) SPEI-12; c) spring; d) summer; e) fall; f) winter.**

### 3.2 Characteristics of spatial and temporal variations of NPP

NPP estimation faces challenges from model structure, input data, and spatiotemporal variations, potentially introducing errors [43]. Here, CASA model accuracy for 2001–2020 North China's humid and semi-humid warm temperate regions NPP is constrained by input data quality, where 1 km-resolution remote sensing data may cause bias. However, based on the annual averages of different land use types in the study area, the validation of MOD17A3 data (Fig 5a) confirms that the outputs generated by CASA are consistent with the baseline data ($R^2 = 0.8766$, $p < 0.01$), thus supporting the reliability of these regional estimates.

NPP is increasing at an average rate of 5.35g $C·m^{-2}a^{-1}$ (Fig 5b). In the past 20 years, the variation range has been between 399.39g $C·m^{-2}a^{-1}$ and 544.23g $C·m^{-2}a^{-1}$, with an average vegetation NPP of 493.53g $C·m^{-2}a^{-1}$. Overall, the yearly average NPP for all terrain types is increasing. Grassland has the highest annual average growth rate (8.02g $C·m^{-2}a^{-1}$), followed by forest land (4.03g $C·m^{-2}a^{-1}$).

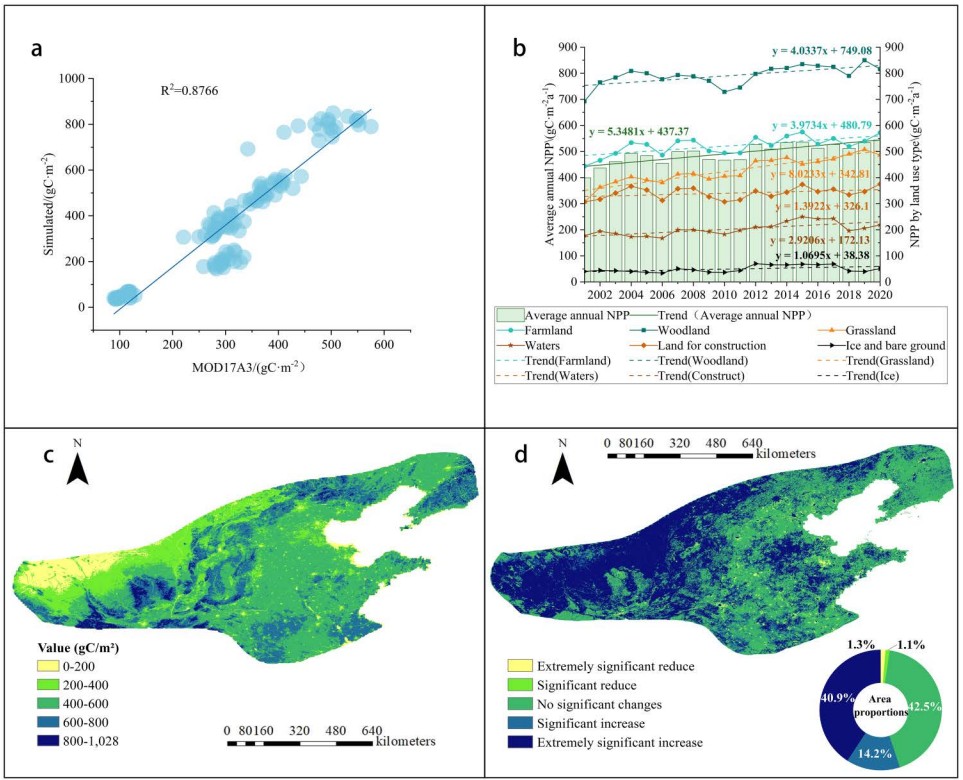

**Fig 5.** a)Validation of MODIS remotely sensed NPP data against CASA modeled NPP data;b)Average annual changes in NPP and NPP for each type of land use, 2001-2020; c) Average NPP from 2001 to 2020; d) Significant changing trends of NPP.

Most of the yearly average NPP value ranged from 200 to 800 g C/m², covering 89.63% of the region. Spatially, vegetation NPP exhibited a progressive decline from central-northern zones toward eastern and western peripheries (Fig 5c). High-value clusters concentrated along the Taihang, Luliang, and Liupan Mountains, corresponding predominantly to forest ecosystems. Western low-value areas featured grassland mosaics with fragmented cropland, while eastern low-value sectors were dominated by contiguous cultivated land. The majority of the vegetation NPP in this region has increased significantly over the last two decades, accounting for 55.10% of the total area (Fig 5d), with a considerable drop accounting for only 2.4%.

### 3.3. Factors affecting the influence of NPP

**3.3.1. Analysis of the impact of climatic factors on NPP.** Precipitation was the main driving factor (Fig 6). Precipitation exhibited a mean correlation coefficient of 0.25 with NPP across the study area, with 86.79% of regions showing positive covariation. Statistically significant correlations ($p < 0.05$) were detected in 21.27% of the territory, primarily clustered within western and northern grassland-cropland mosaics. Temperature and solar radiation demonstrated weaker influences, with respective NPP correlation coefficients of 0.09 (7.01% significant area) and −0.20 (0.22% significant area). Crucially, vegetation NPP exhibited the strongest coupling with mean annual precipitation across North China's humid and semi-humid warm temperate regions when compared to other climate drivers.

The partial correlation analysis confirms precipitation's vital function [44–45]. Controlling for confounding variables, the partial correlation between precipitation and NPP ($r = 0.24$ demonstrated statistical significance ($p < 0.05$) across 22.63%

**Fig 6. Correlation analysis between meteorological factors and NPP (correlation coefficients in the upper part and partial correlation coefficients in the lower part).**

of the study area—exceeding the significant areas for temperature (r = 0.17; 16.20% significant area) and solar radiation (r = −0.17, 0.43% significant area).

**3.3.2. Analysis of the impact of drought on NPP.** On a monthly scale, SPEI-1 showed a positive correlation with NPP in 90.93% of cases, with only 8.21% being statistically significant. Meanwhile, SSMI-1 exhibited positive correlations in 85.56% of cases, but only 0.01% were statistically significant. Both findings indicate a relatively limited association between vegetation and drought (Fig 7). At the annual scale, the proportion of positive correlations for SPEI-12 decreased to 80.16%, but the proportion of significant correlations increased to 16.5%, with highly correlated areas concentrated in grasslands. The proportion of positive correlations for SSMI-12 further decreased to 68.23%, while the proportion of significant correlations increased to 30.05%. The spatial pattern resembled that of SPEI-12, indicating that the sensitivity of grassland ecosystems to long-term water accumulation was reflected in both indices. At the seasonal scale, the intensity of positive correlations for SPEI-3 peaked during summer at 0.92, with 27.09% of areas exhibiting statistically significant correlations (higher than the annual scale). SSMI-3 showed positive correlations in 76.02% of areas, with 34.13% being significant, and both indices' high-value zones were distributed across grasslands. Winter exhibited 82.27% negative correlations with SPEI-3 (peak −0.91), while winter SSMI-3 showed only 42.25% negative correlations with low significance. This clearly demonstrates that during cold-limited periods, SPEI—based on evapotranspiration-precipitation balance—provides a more physiologically meaningful representation of vegetation water stress.

Regarding responses across vegetation types (Fig 7), grassland ecosystems exhibited the highest sensitivity and strongest correlations to both drought indices. Cultivated land ecosystems showed moderate sensitivity, while woodland ecosystems demonstrated the weakest response. This reflects differing adaptive strategies and response mechanisms to drought stress across vegetation types. Spring and summer, as the peak growing seasons, represent the most sensitive periods for vegetation responses to drought stress. Both SPEI and SSMI exhibit relatively high correlation strengths and

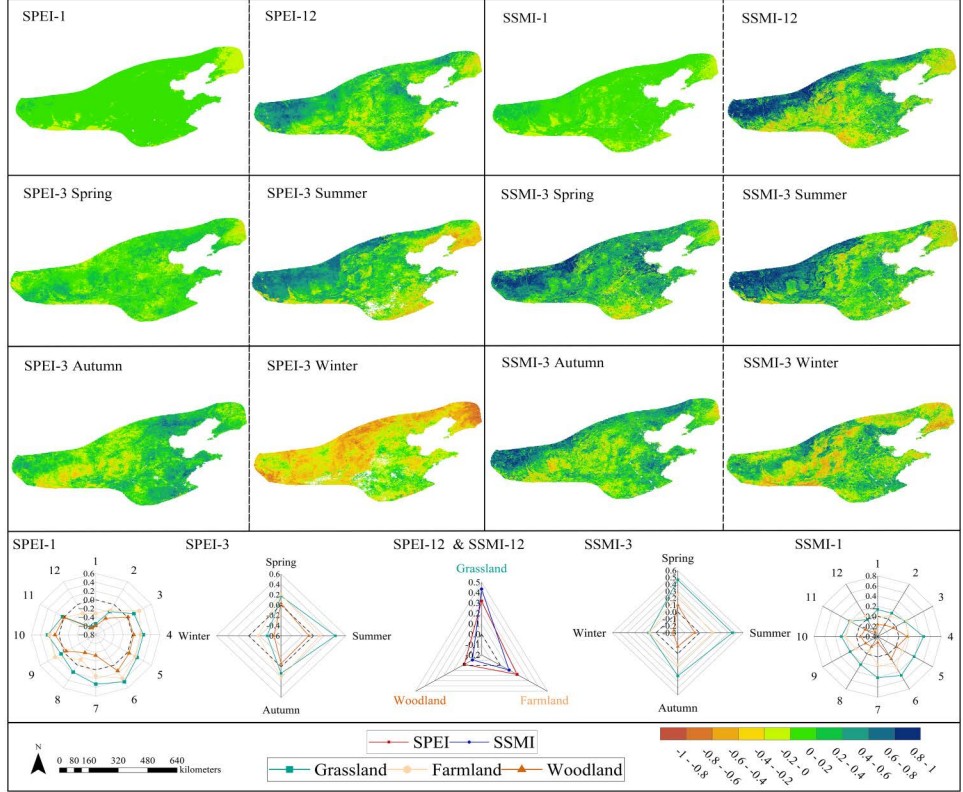

**Fig 7. This section explores the correlation between SPEI and NPP from two complementary perspectives.** (1) The spatial distribution patterns of these correlations across various time scales, including both SPEI and SSMI; (2) For different types of vegetation, mean correlation charts for SPEI and SSMI at the monthly, seasonal, and annual scales are presented to analyze the differences in responses among different ecosystems.

significance levels during these seasons. Winter, however, displays unique correlation patterns, particularly with forest ecosystems generally showing negative correlations during this season.

The overall trends observed in the SPEI-based analysis (Fig 8) align with those from the SSMI-based analysis. Grassland ecosystems exhibited the highest sensitivity, showing a gradual decline in NPP as drought severity increased. In the SSMI-based analysis, grassland NPP changes demonstrated a positive response to drought, though the magnitude of NPP variation was smaller compared to the SPEI-based analysis. Under various drought levels, NPP changes in cropland ecosystems exhibited trends similar to grasslands, though the decline was less pronounced. Croplands also demonstrated characteristics consistent with human cultivation practices: during drought conditions (excluding extreme drought), both cultivation and early crop growth phases showed positive ΔNPP change rates (0%−30%), indicating a growth phase. Analysis based on SSMI drought levels revealed that ΔNPP in croplands consistently exceeded that in grasslands under drought stress. Cropland NPP exhibited lower sensitivity to soil moisture changes than grasslands, primarily attributed to the moderating effects of agricultural management practices. Forest ecosystems demonstrated distinct NPP variation patterns across drought levels compared to both grasslands and croplands. Results based on SPEI drought levels showed consistent trends with those based on SSMI drought levels, though specific magnitude differences existed. Under mild and moderate drought conditions, NPP changes in forest ecosystems were relatively minor. Only under severe and extreme drought conditions did forest NPP exhibit more pronounced declines, though the magnitude of these declines remained lower than that observed in grassland ecosystems.

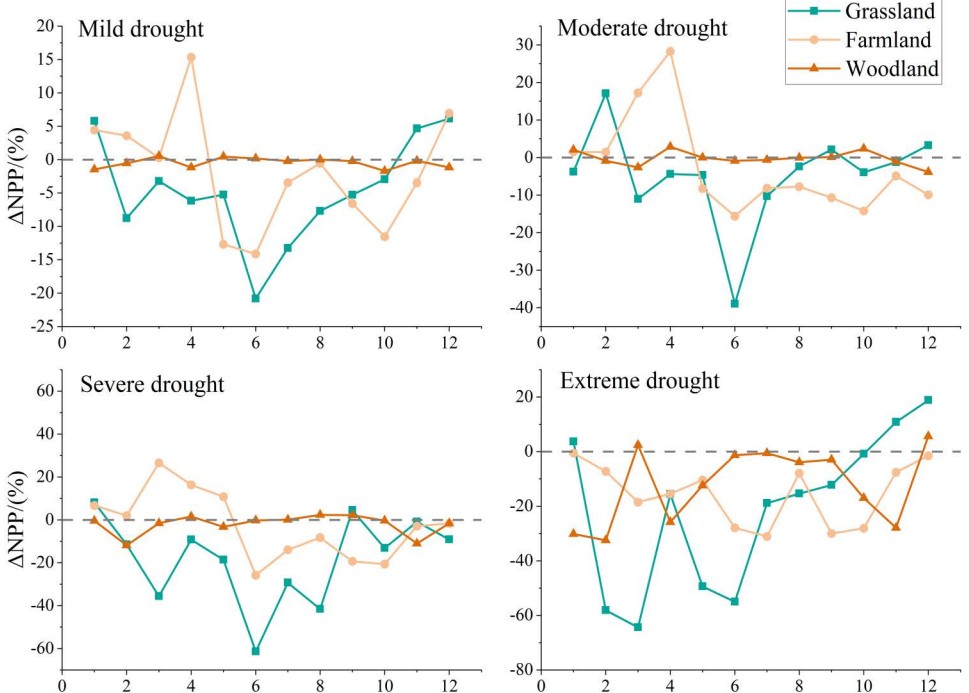

**Fig 8. Based on the SPEI drought classification criteria, different levels of drought were identified, and the changes in NPP for various vegetation types under the influence of these different drought levels were analyzed.** ΔNPP represents the percentage change in NPP relative to the baseline value, which is calculated using the average NPP of vegetation in months without drought across different years.

Under the four drought severity levels classified by SPEI (Fig 8), ΔNPP exhibited negative anomalies across grasslands, croplands, and woodlands during March–May (early growing season), with the magnitude of negative anomalies increasing as drought severity intensified. Within the same severity level, grasslands showed the deepest negative anomalies, while woodlands exhibited the shallowest. From June to August (the peak growing season), the magnitude of negative anomalies for the same vegetation types exceeded that observed from March to May. Results based on SSMI-classified drought levels also showed the same trend. This indicates that drought during the early growing season has a persistent weakening effect on NPP in later stages.

## 4. Discussion

### 4.1. Drought analysis of the study area

The North China region is a typical climatic transition zone, where precipitation and temperature exhibit distinct gradients, resulting in pronounced transitional characteristics in its ecosystems and agricultural production methods [46]. The North China Plain also serves as China's most critical grain production base, with its stable grain output holding strategic significance for national food security [47]. The Northern agro-pastoral transition zone is China's largest typical transition zone, with its North China agricultural region serving as a transitional area between agricultural and livestock systems that is highly sensitive to climate change [48–49]. This study reveals that the humid and semi-humid warm-temperate region of North China, as a climate-sensitive transition zone, exhibits rapid responses in net primary productivity to drought events. This finding aligns with previous research highlighting the sensitivity of vegetation-climate interactions in this area [50]. Furthermore, the observed aridification trend corroborates the pattern of drought intensification in North China documented

by [51]. This study also reveals that both temperature and precipitation are showing an upward trend in the humid and semi-humid warm temperate regions of North China. This finding aligns with the results of Zhao et al. [52], which explicitly indicated a significant increase in precipitation after 2012, occurring concurrently with rising temperatures (warming and moistening of the atmosphere). The study area exhibits a slight decrease in the area affected by drought at both the annual and seasonal scales (except for winter), indicating an overall trend of slight drought alleviation. This phenomenon may be related to the findings of Zhou et al. [53], which suggest that increased precipitation mitigates water limitations exacerbated by vegetation recovery and climate warming, thereby slightly alleviating drought conditions. Drought exhibits temporal-scale divergence: long-term scales show lower intensity due to stable precipitation and water-heat synchronous warming reducing extreme drought risks [54], while short-term scales experience frequent high-intensity droughts from spring precipitation deficits (precipitation from March to May is only 1/2–1/3 of evaporation) and pre-monsoon water shortages [55], aligning with Huang et al.'s regional drought analyses [56].

## 4.2. Characterization of NPP distribution

Over the 2001−2020 period, vegetation NPP across the study region exhibited a general increasing trend, characterized by declining values from central areas toward eastern and western peripheries. This spatial-temporal pattern aligns with the observations of Guo [57]. Primarily climate-driven, increased precipitation enhanced water supply while moderate temperatures extended growing seasons and boosted photosynthetic efficiency, collectively promoting NPP growth [58]. Secondarily, vegetation differentiation manifested as grassland/woodland dominance in central high-NPP areas contrasting with east/west zones constrained by farmland expansion and moisture limitations, reducing growth rates [59]. Additionally, anthropogenic interventions elevated grassland's annual growth ($8.02 gC \cdot m^{-2} a^{-1}$) above woodland ($4.03 gC \cdot m^{-2} a^{-1}$) due to rapid water-heat responses, with ecological restoration (farmland conversion) and irrigation synergistically accelerating plant growth [60].

## 4.3. Analysis of factors affecting NPP

NPP changes are co-regulated by precipitation (dominant), temperature, and solar radiation. Precipitation correlates significantly positively with NPP, especially in semi-humid areas under water-limited conditions, aligning with Yang et al.'s study on climate impacts on China's vegetation carbon sink [61]. Temperature impacts on NPP exhibit regional divergence. In the Guanzhong Plain, suitable precipitation with higher temperatures promotes NPP growth [62]. This positive effect concentrates in high-warming areas like Shandong Province (0.036°C/a) (Table 3), where most zones show significant NPP increases (Fig 5d). Temperature rise may boost NPP via extended growing seasons and enhanced photosynthesis, while solar radiation displays weak negative correlations. Although increased radiation intensifies evapotranspiration and water stress, vegetation's low light sensitivity under abundant radiation minimizes inhibition [63].

Monthly drought-NPP correlations are low due to vegetation's short-term buffering (e.g., root hydraulic redistribution), whereas seasonal/interannual scales show significant correlations from cumulative water integration [64–65]. This study primarily employed SPEI and SSMI as indices to characterize drought impacts on vegetation NPP. First, SPEI, as a meteorological drought index, primarily reflects the balance between atmospheric water supply (precipitation) and demand (potential

**Table 3. The annual average temperature growth rate of each province in the study area.**

| Province | Growth rate (°C/a) | Province | Growth rate (°C/a) | Province | Growth rate (°C/a) |
|---|---|---|---|---|---|
| Shandong | 0.036 | Liaoning | 0.025 | Henan | 0.012 |
| Jiangsu | 0.033 | Anhui | 0.021 | Ningxia | 0.011 |
| Hebei | 0.027 | Shanxi | 0.021 | Gansu | 0.006 |
| Tianjin | 0.026 | Beijing | 0.020 | Shaanxi | 0.001 |

evapotranspiration) [66]. It captures anomalies in atmospheric moisture conditions, serving as an "early warning signal" for vegetation experiencing water stress. Second, SSMI directly reflects actual available soil moisture content, serving as an indicator of water conditions directly accessible to plant roots. It more directly links vegetation physiological processes and productivity [67]. Despite differences between SPEI and SSMI, drought typically begins as meteorological drought, gradually evolves into soil drought, and ultimately impacts vegetation. Combining both indices enables a more comprehensive and accurate revelation of the complex mechanisms by which drought affects vegetation NPP, demonstrating clear synergistic advantages. Afshar et al. in studying the impact of concurrent droughts on global NPP losses, found that significant NPP declines were often accompanied by simultaneous reductions in both SPEI and SSMI, underscoring the critical importance of integrated monitoring of atmospheric and soil drought for assessing ecological risks [68]. This study also demonstrates the synergistic advantages of SPEI and SSMI in more effectively analyzing net primary productivity responses to drought across different vegetation types. Both the correlation between SPEI and grassland NPP and the correlation between SSMI and NPP were higher than those for other vegetation types (Fig 7). Additionally, changes in grassland NPP were the most pronounced across different drought severity levels (Fig 8). This result aligns with findings by Lyu et al. [69], who demonstrated that grassland vegetation, characterized by shallow and densely distributed roots, exhibits heightened sensitivity to surface water availability. Consequently, grasslands demonstrate stronger synchrony across various drought events compared to other vegetation types. In correlation analyses with drought indices (SPEI and SSMI), forest NPP exhibited lower correlation coefficients than other vegetation types. Under varying drought severity levels, forest NPP showed relatively minor fluctuations, with significant changes occurring only under extreme drought conditions. This further demonstrates that trees possess a degree of resistance to environmental changes and exhibit a certain degree of lag in their response to such changes. This conclusion aligns with the findings of Xiao et al. [70]. This study demonstrates that by comparing SPEI and SSMI with NPP across specific vegetation types, more reliable assessment results can be obtained.

This study primarily examines the impact of drought on net primary productivity (NPP) in vegetation. Results indicate that drought events significantly suppress vegetation productivity, with varying response levels across different vegetation types. Notably, human activities constitute a critical factor influencing NPP dynamics and must be fully considered in ecological assessments [71]. In the mountainous regions of Shandong, land-use changes accounted for 68.4% of the variation in net primary productivity [72], while the establishment of nature reserves enhanced regional ecosystem services by 80% [73]. However, spring-planted crops remain vulnerable to short-term drought due to their shallow root systems [74]; long-term irrigation effectively mitigates drought stress—demonstrating the spatiotemporal impacts of human activities on vegetation-climate interactions. As this study focuses on natural mechanisms, the quantitative role of human activities remains unexamined. Future research should integrate the coupled effects of natural and anthropogenic factors, using multi-source data fusion and model simulations to discern the relative contributions of different drivers, thereby providing a scientific basis for sustainable ecosystem management.

## 5. Conclusion

This study employed SPEI to characterize drought and CASA-modeled NPP (2001–2020) in the North China's humid and semi-humid warm temperate regions, analyzing spatiotemporal drought-NPP dynamics and drought impacts.

(1) The climate-sensitive transition zones of North China experienced warming-wetting trends where synchronous temperature-precipitation rises slightly alleviated drought overall. Stable precipitation variability and water-heat synchronization maintained low long-term drought intensity, but frequent high-intensity droughts occurred at short-term scales.

(2) Vegetation NPP increased overall (55.10% area with significant growth), spatially declining from center to east and west. Grassland showed peak annual growth ($8.02 gC·m^{-2}a^{-1}$), then woodland ($4.03 gC·m^{-2}a^{-1}$).

(3) Precipitation dominates vegetation NPP changes; temperature rises promote NPP accumulation but exhibit geographic variation. In the climate-sensitive transition zones of North China, abundant sunlight induces low radiation sensitivity, making temperature and radiation less influential than precipitation.

(4) Vegetation exhibits distinct mechanisms in responding to drought: grasslands show the most pronounced reaction due to their high synchrony, while forests demonstrate drought resistance and delayed responses. In terms of temporal distribution, 80.16% of the regions (primarily grasslands) exhibited a positive correlation with the SPEI-12 index on an annual scale. The SPEI-3 index reached a maximum of 0.92 during summer, while the winter value was −0.91. This correlation was markedly stronger than the weak association indicated by the monthly SPEI-1.

## Supporting information

**S1 Table. Changes in the area of dry and wet regions.**
(DOCX)

**S1 Fig. Compare and verify historical drought episodes with the drought results obtained using the SPEI index during the same periods.** Additionally, calculate the Probability of Detection (POD), False Alarm Ratio (FAR), and Critical Success Index (CSI) for these historical drought events and the SPEI-based drought estimates, in order to assess the accuracy of the SPEI index. (The orange dashed lines indicate the historical drought periods: the spring drought in 2004; the spring and summer drought in 2005; the spring and autumn drought in 2006; the winter, spring, and summer drought from November 2008–2009; the summer and autumn drought in 2010; the spring drought in 2013; the winter, spring, and summer drought from December 2013–2014; the winter, spring, and summer drought from November 2014–2015; and the consecutive autumn and winter droughts from October 2017–2018.).
(TIF)

**S2 Fig. Based on the SSMI drought classification criteria, different levels of drought were identified, and the changes in NPP for various vegetation types under the influence of these different drought levels were analyzed.** ΔNPP represents the percentage change in NPP relative to the baseline value, which is calculated using the average NPP of vegetation in months without drought across different years.
(TIF)

## Author contributions

**Conceptualization:** Xueying Zhang, Bo Zhang, Xiao Hou.

**Formal analysis:** Xueying Zhang, Xiao Hou, Di An.

**Methodology:** Xueying Zhang, Bo Zhang.

**Software:** Xueying Zhang, Zhexi Wu.

**Validation:** Xueying Zhang, Xiao Hou.

**Visualization:** Xueying Zhang, Di An, Zhexi Wu.

**Writing – original draft:** Xueying Zhang.

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
