## [Decision Letter · Decision Letter 0]

12 Nov 2025

Dear Dr. Zhang,

Thank you for submitting your manuscript to PLOS ONE. After careful consideration, we feel that it has merit but does not fully meet PLOS ONE’s publication criteria as it currently stands. Therefore, we invite you to submit a revised version of the manuscript that addresses the points raised during the review process.

**ACADEMIC EDITOR:** I have received comments from both reviewers; each recommends revision, and I concur. Please prepare a careful, point-by-point response that addresses every reviewer comment.

We look forward to receiving your revised manuscript.

Kind regards,

Bao Yang, Ph.D, Prof.

Academic Editor

PLOS ONE

Journal Requirements:

This research was funded by the National Natural Science Foundation of China (41561024)

3. We note that Figure(s) 1, 4, 5, 6, 7 in your submission contain [map/satellite] images which may be copyrighted. All PLOS content is published under the Creative Commons Attribution License (CC BY 4.0), which means that the manuscript, images, and Supporting Information files will be freely available online, and any third party is permitted to access, download, copy, distribute, and use these materials in any way, even commercially, with proper attribution. For these reasons, we cannot publish previously copyrighted maps or satellite images created using proprietary data, such as Google software (Google Maps, Street View, and Earth). For more information, see our copyright guidelines: http://journals.plos.org/plosone/s/licenses-and-copyright.

a. You may seek permission from the original copyright holder of Figure(s) 1, 4, 5, 6, 7 to publish the content specifically under the CC BY 4.0 license.

4. We notice that your supplementary figures are uploaded with the file type 'Figure'. Please amend the file type to 'Supporting Information'. Please ensure that each Supporting Information file has a legend listed in the manuscript after the references list.

5. Please remove your figures from within your manuscript file, leaving only the individual TIFF/EPS image files, uploaded separately. These will be automatically included in the reviewers’ PDF**.**

Reviewers' comments:

Reviewer's Responses to Questions

**Comments to the Author**

1. Is the manuscript technically sound, and do the data support the conclusions?

Reviewer #1: Yes

Reviewer #2: Yes

2. Has the statistical analysis been performed appropriately and rigorously?

Reviewer #1: Yes

Reviewer #2: Yes

3. Have the authors made all data underlying the findings in their manuscript fully available?

Reviewer #1: Yes

Reviewer #2: Yes

4. Is the manuscript presented in an intelligible fashion and written in standard English?

Reviewer #1: Yes

Reviewer #2: Yes

Reviewer #1: The manuscript analyzes drought characteristics at multiple temporal scales (2001-2020) using the Standardized Potential Evapotranspiration Index (SPEI) and Standardized Soil Moisture Index (SSMI). The research topic is of clear practical importance, particularly for understanding regional ecosystem carbon cycles, drought-response mechanisms, and ecological restoration under climate-change scenarios. However, several shortcomings remain in the current version of the manuscript.

1.The manuscript contains multiple instances of writing errors and the author must carefully review it. For example, in sections such as the abstract and keywords, "climate-sensitive" is incorrectly written as "limate-sensitive." Additionally, a multiplication sign (×) is missing in Equation (2), and there is a error in line 314 with "figd."

2. In the data processing methods section, the author used DFI to calculate drought frequency. The calculation formula for DFI should be provided, clarifying which drought index was used for the computation.

3. The manuscript divides the drought class according to the SSMI index(Table 2), but a significant disconnect emerges in the subsequent core analysis: SSMI is only used to validate interannual trend consistency with SPEI-12. The manuscript fails to quantify the impacts of different SSMI drought levels on vegetation NPP based on this classification standard. Furthermore, throughout the analysis of drought effects on NPP, only SPEI is employed as the drought indicator, with no incorporation of the SSMI classification into the evaluation. This renders the design of the SSMI metric redundant, creates an inconsistency with the multi-indicator approach stated in the abstract, and weakens the overall integrity of the research.

4. In section 3.1 of the results and analysis, only the changing trends of SPEI-12 and SSMI are presented graphically. Although the text states that "the trajectories of the two changes are nearly identical," no correlation or significance test results are provided. It is recommended that the authors include a correlation analysis between SPEI and SSMI to quantitatively validate their consistency, thereby enhancing the scientific rigor and persuasiveness of the conclusion.

5. Since the manuscript lists the SSMI and concludes that " grasslands show higher drought correlations than other types due to shallow, dense roots sensitive to surface moisture, with strong event synchrony," it would be beneficial to supplement this with a "comparison of the correlations between NPP and SSMI across different vegetation types (grassland/woodland/farmland)." This addition would enhance the persuasiveness and scientific value of the study.

6. The article uses SPEI-1, SPEI-3, and SPEI-12 to represent different drought timescales, but the data processing methods section does not explain what these specific indices represent or how they are calculated.

7. The manuscript currently combines results description with mechanistic explanations in Section 3.3.2 ("Analysis of the impact of drought on NPP"). For instance, the statement: " Grasslands show higher drought correlations than other types due to shallow, dense roots sensitive to surface moisture, with strong event synchrony" Such causal explanations belong to ecological mechanism inferences and should be placed in the Discussion section rather than in Results. The Results section should solely report factual findings, while the Discussion should interpret the underlying reasons and connect them to relevant research papers.

8.Figure 7 claims that NPP and SPEI show a weak correlation at the monthly scale but a significantly positive correlation at the annual scale. However, the third radar chart in Figure 7—purportedly showing annual-scale correlation coefficients—still displays distinct correlation values for each month (January to December). This suggests that the so-called "annual-scale" analysis was likely performed by correlating monthly NPP values with SPEI-12, rather than using annual aggregated NPP data with SPEI-12. Such an approach is statistically inappropriate, and the authors should provide a clarification on this issue.

9. In Section 4.2 of the Discussion, the authors state: " Human activities constitute significant drivers of NPP dynamics and necessitate explicit consideration in ecological assessments." However, the entire research framework is confined to meteorological variables and does not incorporate any human activity-related data or analysis, which undermines the logical coherence of the manuscript. It is recommended that the authors reframe this statement as a "future research prospect," clarifying that human activities could be a potential driving factor to be considered in future studies rather than being presented as part of the current findings.

10. The current discussion remains insufficiently in-depth and lacks specificity. It is necessary to build on the research findings presented in the manuscript, supplemented by extensive references to relevant peer-reviewed literature, to deepen the understanding of the climate-sensitive transition zones of North China' uniqueness and develop concrete inferences.

11. Figure S1 presents a "validation" that is more of a visual comparison. There are significant mismatches with historical drought records across multiple years, rendering the conclusion of "demonstrating its applicability for drought monitoring" insufficient and unsubstantiated.

12. Figure 2 suffers from poor readability.

Firstly, the drought severity classifications ("Mild drought" "Moderate drought" and "Severe drought") shown in the figure are not based on standard SPEI threshold criteria. For instance, in the panel labeled "Severe Drought," some SPEI values remain above -0.5, which does not constitute a drought condition according to established classification systems.

Secondly, the comparison is confounded by the fact that the data for different vegetation types correspond to entirely different time periods. Consequently, the observed differences in vegetation response are likely contaminated by interannual climatic variability and other non-drought factors, making it impossible to isolate the specific impact of drought on NPP.

To improve the interpretability of the figure, it is recommended that the response curves for different land cover types be plotted using distinct colors.

13. The presentation of data sources in the manuscript is inconsistent. It is recommended to standardize the formatting of data citations throughout the article, avoiding non-standard forms such as "from [26-27]".

14. The manuscript contains numerous issues with its figures. The resolution of all images is too low, making text and details difficult to distinguish.

In Figure 1, the scale bars are inconsistently labeled as "kilometers" and "Kilometers".

Figures 2 and 3 have incomplete borders in their legends.

Figure 3 is confusing in its layout. The four spatial distribution maps in the middle, marked with years, are actually expanded from subplot a. However, their current arrangement alongside the main figure creates an unclear information hierarchy, making it difficult to recognize them as elaborations of subplot a. The authors should reconsider the overall layout design.

Figure 4 lacks a title for its legend, and "spei" should be written in its standard form as "SPEI".

Subplots c and d in Figure 5 should each include a north arrow and a scale bar.

Figure 6 lacks both a north arrow and a scale bar. Furthermore, while the caption describes it as showing "correlation coefficients," the figure itself only visually represents correlations without displaying the actual numerical values.

Reviewer #2: Under global warming, the response of ecosystems to drought is also changing. This paper takes the climate-sensitive transition zone of North China (humid-semi-humid warm temperate region) as the study area, and uses the standardized precipitation evapotranspiration index (SPEI) and standardized soil moisture index (SSMI) to analyze drought characteristics at multiple timescales from 2001 to 2020. Combined with the CASA model to simulate net primary productivity (NPP) of vegetation, the impact of drought on NPP is explored. The findings show that drought showed a slight mitigation trend during the study period, while NPP increased overall. Precipitation was the key influencing factor. Grassland NPP was the most sensitive to drought response, while forest NPP showed a lag. The results have some reference value for guiding ecosystem management under warming conditions. I agree to accept it after minor revisions.

Major concerns:

1. Extreme droughts have a significant impact on ecosystems. The paper defines drought events of different degrees. It is suggested that future research could further analyze the impact of major drought events and their frequency during the growing season and even the early growing season on NPP.

2. The authors still use SPEI when evaluating the relationship between environmental factors and farmland. Why not use SSMI? Please note that irrigation has a significant impact on mitigating major droughts in farmland. It is recommended that the authors re-analyze using SSMI, even if the results are not significantly different. This insignificant difference may be due to low resolution (1 km), which makes it impossible to identify soil moisture changes caused by irrigation.

Minors

1. What is Figd in Line 314?

2. Please unify the caption format for figures and tables in the document..

3. Fig. 5: The CASA simulation results and MODIS NPP product results are good, but the sample size is small, and it is unclear whether all land cover types are covered or their distribution. The authors are requested to provide a detailed description of the validation design and increase the validation sample size.

Figure:

1. Fig. 6: The font size for the correlation data is too small; please redraw it.

2. Fig. S2: The authors listed drought conditions for three land cover types under different drought levels. To better support the authors' conclusions in L262-273 of the paper, it is suggested to use an overlay analysis method, overlaying major, moderate, and mild drought events within the study period. The results of this analysis will be more valuable.

**Do you want your identity to be public for this peer review?** For information about this choice, including consent withdrawal, please see our Privacy Policy

Reviewer #1: No

Reviewer #2: No

---

## [Author Response · Author response to Decision Letter 1]

7 Jan 2026

Comments from the Editor:

Response: We thank all the editors and reviewers for their valuable comments and suggestions. We have carefully revised the manuscript to enhance its clarity and facilitate the understanding of the readers. Our point-to-point responses are presented below. We hope that the revision would satisfactorily address the comments and concerns of the editors and reviewers.

Regarding the journal requirements, the following modifications have been made:

1.I have ensured that my manuscript complies with the formatting requirements of PLOS ONE.

2.Funding information has been explicitly stated within the manuscript.

3.Firstly, the website addresses from which the data were obtained have been provided in the data description section of the submission system and can be accessed through these links. Secondly, the map of the study area was created using open-source software, based on the “Comprehensive Natural Geographic Division” established by Mr. Zhao Songqiao (relevant literature has been cited). The data website address is: “https://osgeo.cn/map/m0148/”. This data website is supported by the Open Source Geospatial Foundation, a non-profit organization whose main goal is to promote the collaborative development of open-source geospatial technologies and data. There are no copyright issues regarding these data. Additionally, in the manuscript, the relevant website addresses for the data sources (such as temperature, precipitation, solar radiation, SPEI datasets, NPP datasets, and land use type datasets) have been listed, and references have been provided. These data are publicly available, and their use in scientific research papers, with proper citation, is completely permissible and does not involve any copyright issues.Due to the issue of the map website being inaccessible, a screenshot of the webpage has been included in the Response to Reviewers.

4.All other upload issues have been strictly addressed according to the requirements.

Reviewer#1

Response: Thanks for the positive comments.

Comment1�The manuscript contains multiple instances of writing errors and the author must carefully review it. For example, in sections such as the abstract and keywords, "climate-sensitive" is incorrectly written as "limate-sensitive." Additionally, a multiplication sign (×) is missing in Equation (2), and there is a error in line 314 with "figd."

Response: Thank you for providing such detailed suggestions. I have carefully proofread and revised the manuscript, correcting errors in spelling, phrasing, and symbols.

Comment2� In the data processing methods section, the author used DFI to calculate drought frequency. The calculation formula for DFI should be provided, clarifying which drought index was used for the computation.

Response: The formula for the Drought Frequency Index (DFI) has been supplemented with additional explanations. Beyond formula revisions, the use of SPEI for drought identification has been clarified. The introduction section explicitly states that SPEI is primarily employed for overall drought identification, while SSMI is mainly used to identify drought through soil moisture. Therefore, SSMI is combined with SPEI to explore the impact of drought on vegetation NPP.

Please see below for the details:

“ The drought frequency index (DFI) measures the ratio of drought months to total months during the study period. Higher DFI values indicate greater drought frequency [35]. When SPEI ≤ -1, it is considered that a “drought” has occurred. The formula for calculating the frequency of droughts is given in Equation (3):

(3)

Where DFI represents the frequency of droughts; m denotes the number of months during which droughts occur; M represents the total number of months within this time scale. ”( Page 9, Lines 178 - 183 )

“ The Standardized Soil Moisture Index (SSMI) is an agricultural drought monitoring indicator constructed based on historical soil moisture time series. This method not only features a relatively straightforward calculation process but also accounts for the distributional characteristics of soil moisture data. Research on its applicability in regional agricultural drought monitoring provides a scientific basis for establishing operational drought monitoring systems and assessing their impacts [36]. SPEI can provide early warnings of climate risks associated with water deficits, while SSMI can verify whether drought has already caused substantive impacts on surface ecosystems and agricultural production. The two indices form a highly complementary relationship in terms of physical mechanisms, spatiotemporal attributes, and monitoring targets. The combined use of SPEI and SSMI overcomes the limitations of individual indices, providing a more systematic, multidimensional, and reliable scientific framework for drought early warning, precise assessment, process tracking, and impact analysis. Therefore, this study employs the SPEI index to analyze drought characteristics from 2001 to 2020, assess their impact on NPP, and utilize the SSMI to validate the substantive effects of drought on NPP in terrestrial ecosystems. ” ( Pages 3-4, Lines 59 - 72 )

References Zhou, H.K., Wu, J.J., Li, X.H., et al. Suitability of assimilated data-based standardized soil moisture index for agricultural drought monitoring. Acta Ecol Sin. 2019;39:2191–2202. https://doi.org/10.5846/stxb201801190153.

Comment3 : The manuscript divides the drought class according to the SSMI index(Table 2), but a significant disconnect emerges in the subsequent core analysis: SSMI is only used to validate interannual trend consistency with SPEI-12. The manuscript fails to quantify the impacts of different SSMI drought levels on vegetation NPP based on�this classification standard. Furthermore, throughout the analysis of drought effects on NPP, only SPEI is employed as the drought indicator, with no incorporation of the SSMI classification into the evaluation. This renders the design of the SSMI metric redundant, creates an inconsistency with the multi-indicator approach stated in the abstract, and weakens the overall integrity of the research.

Response: Based on your feedback, I agree that this issue has indeed arisen. Consequently, I have incorporated an analysis of the correlation between SSMI and vegetation NPP into my assessment of drought impacts on vegetation NPP. Additionally, I calculated changes in vegetation NPP across different drought severity levels defined by SSMI. By combining analyses of vegetation NPP impacts using both SPEI and SSMI drought indices, I have obtained more scientifically robust results.

Please see below for the details: “ On a monthly scale, SPEI-1 showed a positive correlation with NPP in 90.93% of cases, with only 8.21% being statistically significant. Meanwhile, SSMI-1 exhibited positive correlations in 85.56% of cases, but only 0.01% were statistically significant. Both findings indicate a relatively limited association between vegetation and drought (Fig 7). At the annual scale, the proportion of positive correlations for SPEI-12 decreased to 80.16%, but the proportion of significant correlations increased to 16.5%, with highly correlated areas concentrated in grasslands. The proportion of positive correlations for SSMI-12 further decreased to 68.23%, while the proportion of significant correlations increased to 30.05%. The spatial pattern resembled that of SPEI-12, indicating that the sensitivity of grassland ecosystems to long-term water accumulation was reflected in both indices. At the seasonal scale, the intensity of positive correlations for SPEI-3 peaked during summer at 0.92, with 27.09% of areas exhibiting statistically significant correlations (higher than the annual scale). SSMI-3 showed positive correlations in 76.02% of areas, with 34.13% being significant, and both indices' high-value zones were distributed across grasslands. Winter exhibited 82.27% negative correlations with SPEI-3 (peak −0.91), while winter SSMI-3 showed only 42.25% negative correlations with low significance. This clearly demonstrates that during cold-limited periods, SPEI—based on evapotranspiration-precipitation balance—provides a more physiologically meaningful representation of vegetation water stress.

Regarding responses across vegetation types (Fig 7), grassland ecosystems exhibited the highest sensitivity and strongest correlations to both drought indices. Cultivated land ecosystems showed moderate sensitivity, while woodland ecosystems demonstrated the weakest response. This reflects differing adaptive strategies and response mechanisms to drought stress across vegetation types. Spring and summer, as the peak growing seasons, represent the most sensitive periods for vegetation responses to drought stress. Both SPEI and SSMI exhibit relatively high correlation strengths and significance levels during these seasons. Winter, however, displays unique correlation patterns, particularly with forest ecosystems generally showing negative correlations during this season.

The overall trends observed in the SPEI-based analysis (Fig 8) align with those from the SSMI-based analysis (S2 Fig). Grassland ecosystems exhibited the highest sensitivity, showing a gradual decline in NPP as drought severity increased. In the SSMI-based analysis, grassland NPP changes demonstrated a positive response to drought, though the magnitude of NPP variation was smaller compared to the SPEI-based analysis. Under various drought levels, NPP changes in cropland ecosystems exhibited trends similar to grasslands, though the decline was less pronounced. Croplands also demonstrated characteristics consistent with human cultivation practices: during drought conditions (excluding extreme drought), both cultivation and early crop growth phases showed positive ΔNPP change rates (0%-30%), indicating a growth phase. Analysis based on SSMI drought levels revealed that ΔNPP in croplands consistently exceeded that in grasslands under drought stress. Cropland NPP exhibited lower sensitivity to soil moisture changes than grasslands, primarily attributed to the moderating effects of agricultural management practices. Forest ecosystems demonstrated distinct NPP variation patterns across drought levels compared to both grasslands and croplands. Results based on SPEI drought levels showed consistent trends with those based on SSMI drought levels, though specific magnitude differences existed. Under mild and moderate drought conditions, NPP changes in forest ecosystems were relatively minor. Only under severe and extreme drought conditions did forest NPP exhibit more pronounced declines, though the magnitude of these declines remained lower than that observed in grassland ecosystems. ” ( Pages 14-16, Lines 279 -325 )

Comment4� In section 3.1 of the results and analysis, only the changing trends of SPEI-12 and SSMI are presented graphically. Although the text states that "the trajectories of the two changes are nearly identical," no correlation or significance test results are provided. It is recommended that the authors include a correlation analysis between SPEI and SSMI to quantitatively validate their consistency, thereby enhancing the scientific rigor and persuasiveness of the conclusion.

Response: Based on your suggestions, I conducted a correlation analysis and significance test between SPEI and SSMI. The results indicate that the two indices exhibit a positive correlation at a rate of 91.21%, with a significant positive correlation of 54.61%. Therefore, the combination of SPEI and SSMI, as well as the validation of SPEI by SSMI, is scientifically sound.

Please see below for the details:

References: “ The correlation between SPEI and SSMI is extremely high (the positive correlation reaches 91.21%, and in 54.61% of the regions, this correlation is statistically significant). ” ( Page 11, Lines 211-213 )

Comment5�Since the manuscript lists the SSMI and concludes that " grasslands show higher drought correlations than other types due to shallow, dense roots sensitive to surface moisture, with strong event synchrony," it would be beneficial to supplement this with a "comparison of the correlations between NPP and SSMI across different vegetation types (grassland/woodland/farmland)." This addition would enhance the persuasiveness and scientific value of the study.

Response: To enhance the scientific rigor of the study, I expanded the correlation analysis between NPP and SSMI to include different vegetation types (grassland, cropland, and woodland). Additionally, I conducted a comparative analysis between the correlations of SPEI with NPP across various vegetation types and the correlations of SSMI with NPP across the same vegetation types.

Please see below for the details:

References: “ Regarding responses across vegetation types (Fig 7), grassland ecosystems exhibited the highest sensitivity and strongest correlations to both drought indices. Cultivated land ecosystems showed moderate sensitivity, while woodland ecosystems demonstrated the weakest response. This reflects differing adaptive strategies and response mechanisms to drought stress across vegetation types. Spring and summer, as the peak growing seasons, represent the most sensitive periods for vegetation responses to drought stress. Both SPEI and SSMI exhibit relatively high correlation strengths and significance levels during these seasons. Winter, however, displays unique correlation patterns, particularly with forest ecosystems generally showing negative correlations during this season. ” ( Page 15, Lines 296-303 )

Comment6�The article uses SPEI-1, SPEI-3, and SPEI-12 to represent different drought timescales, but the data processing methods section does not explain what these specific indices represent or how they are calculated.

Response: The fundamental concepts of SPEI-1, SPEI-3, and SPEI-12 have been supplemented, and the basic calculation methods have also been briefly described.

Please see below for the details: “ SPEI-1 is derived from monthly water balance data that is non-cumulative, fitted to a probability distribution, and standardized. It represents the monthly water balance deficit or surplus. This index measures very short-term, near-real-time meteorological drought conditions. It is primarily controlled by monthly precipitation and temperature (which influence evapotranspiration). SPEI-3 is derived from a three-month sliding cumulative monthly water balance, followed by probability distribution fitting and normalization. It represents the cumulative water balance over the past three months (including the current month). It is a seasonal-scale indicator reflecting seasonal moisture trends. SPEI-12 is derived by sliding and accumulating monthly water balances over 12 months, followed by probability distribution fitting and normalization. It represents the cumulative water balance deficit over the past 12 months (including the current month). As an annual-scale indicator, it reflects long-term hydrometeorological trends. ” (Page 7, Lines 133-143 )

References

Comment7�The manuscript currently combines results description with mechanistic explanations in Section 3.3.2 ("Analysis of the impact of drought on NPP"). For instance, the statement: " Grasslands show higher drought correlations than other types due to shallow, dense roots sensitive to surface moisture, with strong event synchrony" Such causal explanations belong to ecological mechanism inferences and should be placed in the Discussion section rather than in Results. The Results section should solely report factual findings, while the Discussion should interpret the underlying reasons and connect them to relevant research papers.

Response: Based on your feedback, I have removed the ecological mechanism explanation regarding vegetation from the results section and relocated it to the “Discussion” section. Within the discussion, I further analyze the ecological mechanisms underlying this phenomenon by examining the correlation between drought indices (SPEI and SSMI) and NPP, as well as the variation in vegetation NPP across different drought severity levels. This analysis includes comparisons with findings from other studies.

Please see below for the details: “Both the correlation between SPEI and grassland NPP and the correlation between SSMI

---

## [Decision Letter · Decision Letter 1]

4 Feb 2026

Dear Dr. Zhang,

Thank you for submitting your manuscript to PLOS ONE. After careful consideration, we feel that it has merit but does not fully meet PLOS ONE’s publication criteria as it currently stands. Therefore, we invite you to submit a revised version of the manuscript that addresses the points raised during the review process.

**ACADEMIC EDITOR: Please respond to the reviewer's comments and do revision carefully.**

We look forward to receiving your revised manuscript.

Kind regards,

Bao Yang, Ph.D, Prof.

Academic Editor

PLOS One

Journal Requirements:

Additional Editor Comments :

A careful revision is needed.

Reviewers' comments:

Reviewer's Responses to Questions

**Comments to the Author**

Reviewer #1: All comments have been addressed

2. Is the manuscript technically sound, and do the data support the conclusions?

Reviewer #1: Yes

3. Has the statistical analysis been performed appropriately and rigorously?

Reviewer #1: Yes

4. Have the authors made all data underlying the findings in their manuscript fully available?

Reviewer #1: Yes

5. Is the manuscript presented in an intelligible fashion and written in standard English?

Reviewer #1: Yes

Reviewer #1: The author has meticulously revised the manuscript, which has significantly improved the quality of the article. However, there are still a few minor issues that need to be addressed.

1.In the Discussion section 4.1, the manuscript states:"This study uncovers climate-drought coupling in North China's humid and semi-humid warm temperate regions: increased precipitation dominates water balance improvement and meteorological drought reduction. Crucially, precipitation rise offsets temperature-driven evapotranspiration losses, slightly mitigating drought." However, the primary evidence currently provided in the paper is based on correlation analysis. Correlation only indicates a statistical association between variables and cannot directly support mechanistic conclusions such as "offset."

2.The discussion section needs to further strengthen the differences and synergistic advantages between SPEI and SSMI. It is recommended to integrate the article’s findings on the correlations among SPEI, SSMI, and NPP to clarify the differences between the two indices in drought monitoring and NPP impact assessment, thereby justifying the necessity and advantages of the dual-indicator approach.

3.In Figure 1, the spacing between the scale markers 80 and 160 is too small.

**Do you want your identity to be public for this peer review?** For information about this choice, including consent withdrawal, please see our Privacy Policy

Reviewer #1: No

---

## [Author Response · Author response to Decision Letter 2]

8 Feb 2026

Submission ID: PONE-D-25-53876

Title: Drought Characteristics and Their Impact on Vegetation Net Primary Productivity in the Climate-Sensitive Transition Zones of North China

Comments from the Editor:

Response: We extend our sincere gratitude once again to all editors and reviewers for their valuable comments and suggestions. After carefully considering each piece of feedback, we have made corresponding revisions and improvements to the manuscript to further enhance the clarity and readability of the text.

Reviewer#1

Response: Thanks for the positive comments.

Comment1�In the Discussion section 4.1, the manuscript states:"This study uncovers climate-drought coupling in North China's humid and semi-humid warm temperate regions: increased precipitation dominates water balance improvement and meteorological drought reduction. Crucially, precipitation rise offsets temperature-driven evapotranspiration losses, slightly mitigating drought." However, the primary evidence currently provided in the paper is based on correlation analysis. Correlation only indicates a statistical association between variables and cannot directly support mechanistic conclusions such as "offset."

Response: Based on your feedback, I have come to fully recognize that certain expressions in the “ Discussion ” section were inappropriate. My original intent was to analyze the causes behind the trend changes in climatic factors and drought conditions within this study during the discussion phase, but the wording fell short. I have revised the text accordingly, now presenting it in three distinct sections: trends in temperature and precipitation changes, overall trends in drought conditions, and an analysis of the causes.

Please see below for the details: “ This study also reveals that both temperature and precipitation are showing an upward trend in the humid and semi-humid warm temperate regions of North China. This finding aligns with the results of Zhao et al. [52], which explicitly indicated a significant increase in precipitation after 2012, occurring concurrently with rising temperatures (warming and moistening of the atmosphere). The study area exhibits a slight decrease in the area affected by drought at both the annual and seasonal scales (except for winter), indicating an overall trend of slight drought alleviation. This phenomenon may be related to the findings of Zhou et al. [53], which suggest that increased precipitation mitigates water limitations exacerbated by vegetation recovery and climate warming, thereby slightly alleviating drought conditions. ”( Page 17, Lines 351 - 359 )

References

Zhao, J.C., Li, Q.Q., Ding, Y.H., Liu, Y.Y., Tan, G.R., Shen, X.Y., Wu, Q.Y. 2023. Interdecadal increase of summer precipitation in North China in the early 2010s and its association with atmospheric circulation anomalies. Acta Meteorologica Sinica, 81(5):764–775. https://doi.org/10.11676/qxxb2023.20220198

Zhou, J.L., Liu, Q., Liang, L.Q., Yan, D.H., Yang, Y.T., Wang, X., Sun, T., Li, S.Z., Gan, L.Y., Wu, J.F. 2024. Water constraints enhanced by revegetation while alleviated by increased precipitation on China’s water-dominated Loess Plateau. Journal of Hydrology, 640:131731. https://doi.org/10.1016/j.jhydrol.2024.131731.

Comment2�The discussion section needs to further strengthen the differences and synergistic advantages between SPEI and SSMI. It is recommended to integrate the article’s findings on the correlations among SPEI, SSMI, and NPP to clarify the differences between the two indices in drought monitoring and NPP impact assessment, thereby justifying the necessity and advantages of the dual-indicator approach.

Response: Thank you for your constructive suggestions. I believe it is very necessary to add further discussions on the SPEI and SSMI indices in the discussion section. I have included discussions on SPEI and SSMI in Section 4.3, covering three aspects: the main differences and respective advantages of SPEI and SSMI, the advantages of comparing them together, as well as relevant literature support. Additionally, the study compares how different vegetation types’ NPP responds to drought (as measured by SPEI or SSMI). In conclusion, using both SPEI and SSMI indicators in conjunction will make the results more scientifically sound.

Please see below for the details: “This study primarily employed SPEI and SSMI as indices to characterize drought impacts on vegetation NPP. First, SPEI, as a meteorological drought index, primarily reflects the balance between atmospheric water supply (precipitation) and demand (potential evapotranspiration) [66]. It captures anomalies in atmospheric moisture conditions, serving as an “early warning signal” for vegetation experiencing water stress. Second, SSMI directly reflects actual available soil moisture content, serving as an indicator of water conditions directly accessible to plant roots. It more directly links vegetation physiological processes and productivity [67]. Despite differences between SPEI and SSMI, drought typically begins as meteorological drought, gradually evolves into soil drought, and ultimately impacts vegetation. Combining both indices enables a more comprehensive and accurate revelation of the complex mechanisms by which drought affects vegetation NPP, demonstrating clear synergistic advantages. Afshar et al. in studying the impact of concurrent droughts on global NPP losses, found that significant NPP declines were often accompanied by simultaneous reductions in both SPEI and SSMI, underscoring the critical importance of integrated monitoring of atmospheric and soil drought for assessing ecological risks[68]. This study also demonstrates the synergistic advantages of SPEI and SSMI in more effectively analyzing net primary productivity responses to drought across different vegetation types. Both the correlation between SPEI and grassland NPP and the correlation between SSMI and NPP were higher than those for other vegetation types (Fig 7). Additionally, changes in grassland NPP were the most pronounced across different drought severity levels (Fig 8 and S2). This result aligns with findings by Lyu et al. [69], who demonstrated that grassland vegetation, characterized by shallow and densely distributed roots, exhibits heightened sensitivity to surface water availability. Consequently, grasslands demonstrate stronger synchrony across various drought events compared to other vegetation types. In correlation analyses with drought indices (SPEI and SSMI), forest NPP exhibited lower correlation coefficients than other vegetation types. Under varying drought severity levels, forest NPP showed relatively minor fluctuations, with significant changes occurring only under extreme drought conditions. This further demonstrates that trees possess a degree of resistance to environmental changes and exhibit a certain degree of lag in their response to such changes. This conclusion aligns with the findings of Xiao et al. [70]. This study demonstrates that by comparing SPEI and SSMI with NPP across specific vegetation types, more reliable assessment results can be obtained. ” ( Pages 19 - 20, Lines 390 - 418 )

References

Stagge, J.H., Tallaksen, L.M., Xu, C.Y., et al. 2014. Standardized precipitation-evapotranspiration index (SPEI): Sensitivity to potential evapotranspiration model and parameters. In Servat, E., Paturel, J.-E., Dezetter, A., et al. (Eds.), Hydrology in a Changing World: Environmental and Human Dimensions - Proceedings of FRIEND-Water 2014, 363:367–373. Copernicus Publications. https://doi.org/10.5194/piahs-363-367-2014.

Xie, L.L., Li, Y., Zhang, Z.Y., et al. 2025. Exploring the combined effects of drought and drought-flood abrupt alternation on vegetation using interpretable machine learning model and r-vine copula function. Agricultural and Forest Meteorology, 370:110568. https://doi.org/10.1016/j.agrformet.2025.110568.

Afshar, M.H., Bulut, B., Duzenli, E., et al. 2022. Global spatiotemporal consistency between meteorological and soil moisture drought indices. Agricultural and Forest Meteorology, 316:108848. https://doi.org/10.1016/j.agrformet.2022.108848.

Lyu, H., Wang, X., Song, N.P., Chen, J., Zhang, Y.F., Wu, X.D., Yu, D., Yang, X.G., Wang, L., and Chen, L. Response of soil moisture dynamics in four typical herbaceous communities to extreme drought precipitation in the desert steppe. Journal of Soil and Water Conservation, 2023, 37(6): 145-152. https://doi.org/10.13870/j.cnki.stbcxb.2023.06.019.

Xiao, J.Y., Zhang, W.Y., Mu, Y.M., Lyu, L.X. Differences of drought tolerance of the main tree species in Dongling Mountain, Beijing, China as indicated by tree rings. Chin J Appl Ecol. 2021;32:3487–3496. https://doi.org/10.13287/j.1001-9332.202110.001.

Comment3�In Figure 1, the spacing between the scale markers 80 and 160 is too small.

Response: Thank you for your detailed suggestions. I have adjusted the scale in Figure 1.

---

## [Editor Report · Decision Letter 2]

11 Feb 2026

Drought Characteristics and Their Impact on Vegetation Net Primary Productivity in the Climate-Sensitive Transition Zones of North China

PONE-D-25-53876R2

Dear Dr. Zhang,

We’re pleased to inform you that your manuscript has been judged scientifically suitable for publication and will be formally accepted for publication once it meets all outstanding technical requirements.

Kind regards,

Bao Yang, Ph.D, Prof.

Academic Editor

PLOS One
---

## [Editor Report · Acceptance letter]

PONE-D-25-53876R2

PLOS One

Dear Dr. Zhang,

I'm pleased to inform you that your manuscript has been deemed suitable for publication in PLOS One. Congratulations! Your manuscript is now being handed over to our production team.

Kind regards,

on behalf of

Dr. Bao Yang

Academic Editor

PLOS One